# Pulmonary function and six-minute-walk test in patients after recovery from COVID-19: A prospective cohort study

Dararat Eksombatchai[1], Thananya Wongsinin[2]*, Thanyakamol Phongnarudech[2], Kanin Thammavaranucupt[2], Naparat Amornputtisathaporn[1], Somnuek Sungkanuparph[2]

1 Department of Medicine, Faculty of Medicine Ramathibodi Hospital, Mahidol University, Bangkok, Thailand,
2 Chakri Naruebodindra Medical Institute, Faculty of Medicine Ramathibodi Hospital, Mahidol University, Samut Prakan, Thailand

* thananya.won@mahidol.ac.th

## Abstract

### Objectives

To study abnormality of spirometry, six-minute walk distance, and chest radiograph among patients recovered from Coronavirus Disease 2019 (COVID-19).

### Methods and study design

A prospective cohort study was conducted in 87 COVID-19 confirmed cases who recovered and discharged from a medical school hospital in Thailand. At the follow-up visit on day 60 after onset of symptoms, patients underwent an evaluation by spirometry (FVC, FEV1, FEV1/FVC, FEF25-75, and PEF), a six-minute-walk test (6MWT), and a chest radiograph.

### Results

There were 35 men and 52 women, with a mean age of 39.6±11.8 years and the mean body mass index (BMI) was 23.8±4.3 kg/m$^2$. Of all, 45 cases had mild symptoms; 35 had non-severe pneumonia, and 7 had severe pneumonia. Abnormality in spirometry was observed in 15 cases (17.2%), with 8% of restrictive defect and 9.2% of obstructive defect. Among the patients with an abnormal spirometry, the majority of the cases were in the severe pneumonia group (71.4%), compared with 15.6% in the non-severe pneumonia group, and 10.2% in the mild symptom group ($p = 0.001$). The mean six-minute-walk distance (6MWD) in the mild symptom and non-severe pneumonia groups was 538±56.8 and 527.5±53.5 meters, respectively. Although the severe pneumonia group tended to have a shorter mean 6-min walking distance, but this was not statistically significant ($p = 0.118$). Twelve patients (13.8%) had abnormal chest radiographs that showed residual fibrosis. This abnormality was more common in the severe pneumonia group (85.7%) and in others (7.5%) ($p<0.001$).

### Conclusions

Abnormal spirometry was noted in 17.2% of COVID-19 survivors with both restrictive and obstructive defects. Severe COVID-19 pneumonia patients had higher prevalence rates of

**Data Availability Statement:** All relevant data are within the manuscript and its Supporting information files.

**Funding:** The authors received no specific funding for this work.

**Competing interests:** The authors have declared that no competing interests exist.

**Abbreviations:** 6MWD, six-minute-walk distance; 6MWT, six-minute-walk test; BMI, body mass index; COVID-19, Coronavirus Disease 2019; FEF, forced expiratory flow; FEV1, forced expiratory volume in 1 second; FVC, forced vital capacity; IQR, interquartile range; PEF, peak expiratory flow; SARS-CoV-2, Severe Acute Respiratory Syndrome Coronavirus 2; SD, standard deviation; WHO, World Health Organization.

abnormal spirometry and residual fibrosis on the chest radiographs when compared to patients in the mild symptom and non-severe pneumonia groups.

## Introduction

Coronavirus Disease 2019 (COVID-19) is an important emerging infectious disease caused by Severe Acute Respiratory Syndrome Coronavirus 2 (SARS-CoV-2). In December 2019, this virus was identified as the cause of a cluster of pneumonia cases in Wuhan, China [1]. COVID-19 was declared as a pandemic by the World Health Organization (WHO) in March of 2020 [2]. Recently, more than 130 million patients were reported globally, with over 2.8 million deaths [3].

COVID-19 patients may present with a spectrum of symptoms ranging from asymptomatic, mild upper respiratory tract symptoms, to severe pneumonia and multiorgan failure. The lung is the most common organ affected in SARS-CoV-2 infection. The predominant pattern of lung abnormalities during illness is ground-glass opacity. Furthermore, many patients have residual opacity on the chest CT scans, in which the main pattern is ground-glass opacity at the time of discharge [4]. The pathology of the lung in COVID-19 patients includes diffused alveolar damage [5], bronchiolitis, alveolitis and interstitial fibrosis [6]. Thus, patients who are infected with SAR-CoV2 may have a restrictive or obstructive defect on a spirometry during recovery. Previous studies [7, 8] in Severe Acute Respiratory Syndrome (SARS) showed that patients had an abnormal pulmonary function test up to 20% after recovery from SARS. Few studies, mainly in China [9–14] have reported abnormal lung function and six-minute-walk test (6MWT) in patients who were infected with SARS-CoV-2 after recovery. Most of these previous studies were conducted only in patients with pneumonia and those with mild symptoms were not included.

In our study, we aimed to investigate lung function test, 6MWT and chest radiograph in all patients after recovery from confirmed COVID-19 that were admitted to our hospital regardless of the severity of symptoms. We compared spirometric parameters between patients in the mild symptom group, non-severe pneumonia group and severe pneumonia group. Additionally, the correlation between spirometry, 6MWT distance and chest radiograph were analyzed.

## Methods

### Study site, design and patients

A prospective cohort study was conducted in all patients with confirmed COVID-19 who recovered and were discharged from a medical-school hospital, Chakri Naruebodindra Medical Institute, Faculty of Medicine Ramathibodi Hospital, Mahidol University, in Samut Prakan, Thailand. We followed up patients who were over 18 years of age at day 60 after the onset of symptoms.

Diagnosis of COVID-19 was confirmed by detecting SARS-CoV-2 RNA using quantitative RT-PCR amplification of SARS-CoV-2 ORF1AB and N gene fragments (Sansure Biotech Inc, Changsha, PR China). Pneumonia was defined as clinical symptoms of respiratory tract infection with abnormal lung imaging compatible with COVID-19 pneumonia. Severe pneumonia was defined as pneumonia with one of the following criteria: respiratory rate >30 breaths/minute, severe respiratory distress or $SpO_2$ <94% at room air according to the WHO, 2020 [15]. The classification of severity was done during the admission.

At the follow-up visit, patients underwent an evaluation by pulmonary function test, chest radiograph and 6MWT. All patients had provided written informed consent before the evaluation. The study protocol was approved by the Institutional Review Board, Faculty of Medicine Ramathibodi Hospital, Mahidol University (approval no. MURA2020/1550).

## Pulmonary function testing

Each patient underwent a standard spirometry (Model 2700–3; Easy on-PC, NDD Medical Technologies, Zurich Switzerland) for forced expiratory volume in the first second ($FEV_1$), forced vital capacity (FVC), $FEV_1$/FVC ratio, forced expiratory flow (FEF) 25–75% and peak expiratory flow (PEF). We conducted all spirometric measurements according to the standard recommendations of the American Thoracic Society (ATS) [16]. We performed both pre-bronchodilator and post-bronchodilator measurements. The results were expressed as a percentage of predicted values using normal values for the population of Thailand [17].

## Statistical analysis

Continuous variables are presented as mean ± standard deviation (SD) for normally distributed data. Non-normally distributed variables are shown as median (interquartile range [IQR]). Categorical variables are presented as frequency (%). A one-way ANOVA test was used to compare means between the groups and a chi-square test was used to analyze for categorical variables. The differences of non-normally distributed variable between groups are compared using Kruskal-Wallis. Independent sample T-test was used to figure out the correlation between chest radiography with lung function and 6MWD. Statistical significance is accepted at a two-sided p-value <0.050. Statistical analysis was conducted using the IBM Statistical Package for the Social Sciences (SPSS) for Windows, Version 24.0 (IBM Corp, Armonk, NY).

## Results

### Characteristics of study patients

147 Patients were diagnosed and admitted with COVID-19 in Chakri Naruebodindra Medical Institute, Faculty of Medicine Ramathibodi Hospital during the study period. Of all, two patients died due to severe pneumonia and multiorgan failure; 52 patients did not participate in this study secondary to inconvenient long-distance transportation from their hometown; and six patients failed to finish the spirometry test (Fig 1).

A total of 87 patients had been included and completed the tests in the study. There were 35 men and 52 women, with a mean age of 39.6 ± 11.8 years and the mean body mass index (BMI) was 23.8 ± 4.3 kg/m². There were nine active smokers (10.3%) and 18 previous smokers (20.7%). The three most common co-morbidities were hypertension (8%), diabetes mellitus (6.9%), and dyslipidemia (4.6%). Among the patient with the underlying diseases, there were some overlapping of them. The number of the patients with DM and HT, HT and DLP, and DM, HT and DLP were 2 (2.3%), 3 (3.4%), and 1 (1.1%), respectively. None of the patients had underlying chronic lung disease, asthma or chronic obstructive pulmonary disease. Of the 87 patients, 45 (51.7%) had mild symptoms; 35 (40.2%) had non-severe pneumonia and 7 (8%) had severe pneumonia (Table 1).

The patients in the severe pneumonia group had a higher BMI and older age than those in the mild symptom and non-severe pneumonia groups. The mean oxygen saturation on admission was significantly lower in severe pneumonia group compared with the mild symptom and non-severe pneumonia groups. The severe pneumonia group had higher levels of serum lactate

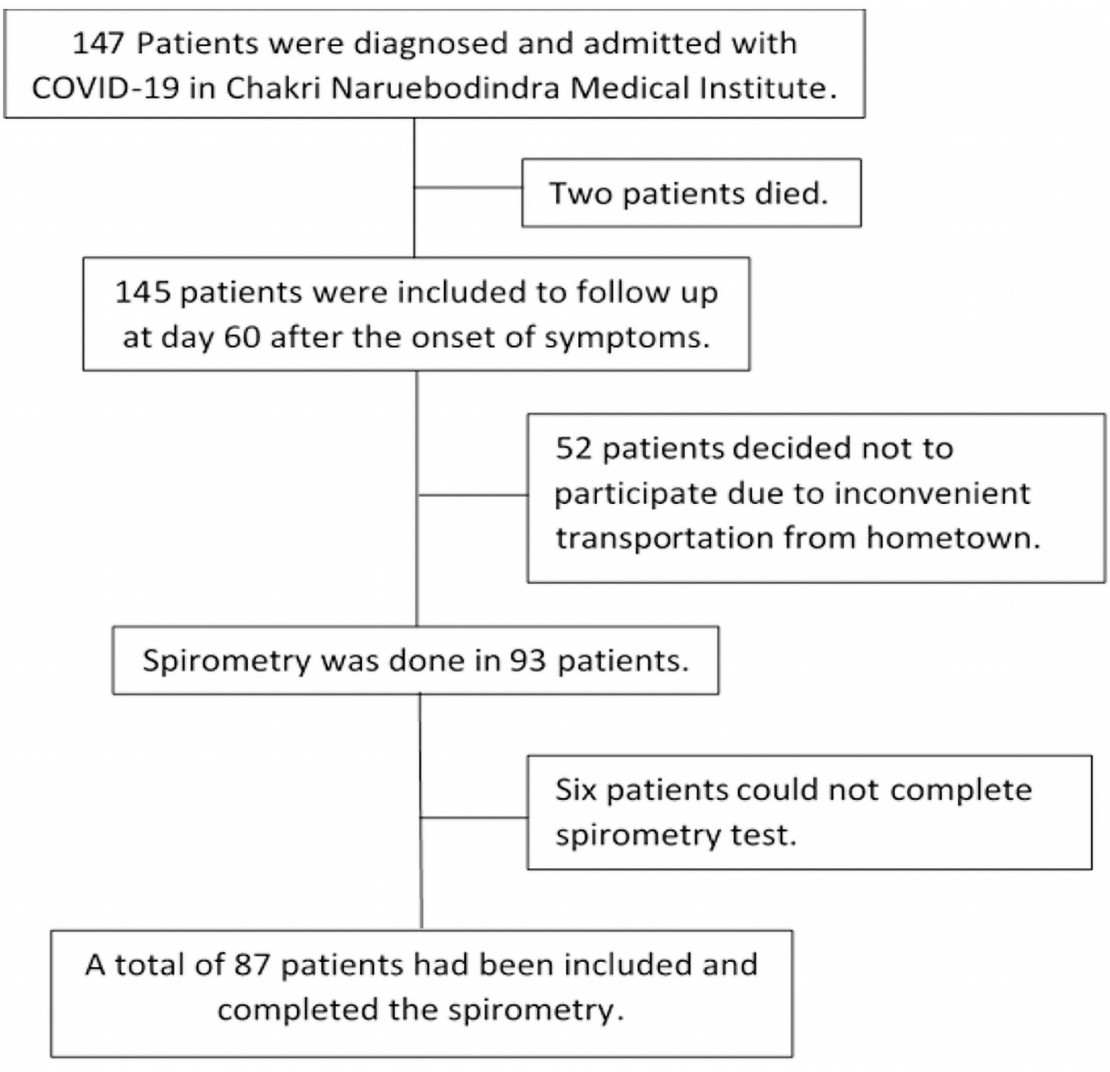

**Fig 1. Flow of the research subjects.**

dehydrogenase (LDH) compared with the mild symptom and non-severe pneumonia groups. This finding was similar to the levels of serum D-dimer that was highest among severe cases. Meanwhile, there were no significant differences in serum creatinine between the three groups ($p = 0.282$). The mean absolute lymphocyte count was lower in the severe pneumonia group during the admission. The average length of stay was significantly higher in severe pneumonia group compared with non-severe pneumonia, and mild symptom group (24.9±5.6 vs 12.9±5.8 and 8.0±2.5 respectively, $p<0.001$) (Table 1).

## Pulmonary function tests

The results of spirometry are shown in Table 2. In all groups, the mean values of FVC, FEV1, FEV1/FVC, FEF25-75 and PEF were within normal ranges. Nevertheless, the mean value of FVC was lower in the severe pneumonia group when compared to the mild symptom and non-severe pneumonia groups in both pre-bronchodilator (84.4% vs 98.1% and 100.4%

**Table 1. Clinical characteristics and laboratory results of 87 recovered COVID-19 patients.**

| Characteristics | Total (N = 87) | Mild symptom (N = 45) | Non-severe pneumonia (N = 35) | Severe pneumonia (N = 7) | P-value |
|---|---|---|---|---|---|
| *Baseline characteristics* | | | | | |
| Age (years), mean ± SD | 39.6 ± 11.8 | 35.9 ± 10.8 | 42.4 ± 11.6 | 50.1 ± 10.2 | 0.020 |
| Male gender, number (%) | 35 (40.2) | 17 (37.8) | 14 (40.0) | 4 (57.1) | 0.640 |
| BMI (kg/m$^2$), mean ± SD | 23.8 ± 4.3 | 22.5 ± 3.7 | 24.9 ± 4.4 | 26.5 ± 4.3 | 0.009 |
| Days of illness prior to admission, (days) mean ± SD | 6.8 ± 3.7 | 7.5 ± 4.0 | 5.7 ± 3.3 | 7.6 ± 2.5 | 0.071 |
| *Previous coexisting disease*, number (%) | | | | | |
| Hypertension | 7 (8) | 2 (4.4) | 2 (5.7) | 3 (42.9) | 0.010 |
| Diabetes mellitus | 6 (6.9) | 0 | 3 (8.6) | 3 (42.9) | 0.001 |
| Dyslipidemia | 4 (4.6) | 1 (2.1) | 1 (3) | 2 (28.6) | 0.031 |
| Coronary artery disease | 1 (1.1) | 0 (0) | 0 (0) | 1 (14.3) | 0.080 |
| *Smoking and alcohol history*, number (%) | | | | | |
| Active smoker | 9 (10.3) | 6 (13.3) | 3 (8.6) | 0 (0) | 0.660 |
| Former smoker | 18 (20.7) | 5 (10.6) | 9 (27.3) | 4 (57.1) | 0.013 |
| Active alcohol drinker | 50 (57.5) | 28 (62.2) | 19 (54.3) | 3 (42.9) | 0.802 |
| Laboratory results | | | | | |
| Oxygen saturation RA (%), mean ± SD | 96.9 ± 2.3 | 97.5 ± 1.2 | 96.9 ± 1.9 | 92.4 ± 4.0 | <0.001 |
| Hemoglobin (g/dl), mean ± SD | 13.5 ± 1.7 | 13.5 ± 2.0 | 13.5 ± 1.1 | 14.0 ± 2.0 | 0.719 |
| White blood cells (cells/mm$^3$), mean ± SD | 5476 ± 1778 | 5554 ± 1644 | 5343 ± 2046 | 5644 ± 1263 | 0.844 |
| ALC (cells/mm$^3$), mean ± SD | 1782 ± 757 | 1952 ± 683 | 1676 ± 847 | 1221 ± 300 | 0.032 |
| Platelet (/mm$^3$), mean ± SD | 234187 ±71723 | 251467 ±55337 | 218409 ±87707 | 202000 ±52783 | 0.056 |
| LDH (U/l), mean ± SD | 197.5 ± 76.6 | 166.9 ± 35.8 | 205.0 ± 59.9 | 357.0 ± 132.1 | <0.001 |
| D-Dimer (ng/ml), median (IQR) | 287 (189–447) | 239 (189–359) | 312 (189–439) | 545 (427–2510) | 0.003 |
| Creatinine (mg/dl), mean ± SD | 0.80 ± 0.20 | 0.80 ± 0.20 | 0.78 ± 0.18 | 0.91 ± 0.35 | 0.282 |
| Length of stay, mean ± SD | 11.3 ± 6.3 | 8.0 ± 2.5 | 12.9 ± 5.8 | 24.9 ± 5.6 | <0.001 |

BMI: body mass index; ALC: Absolute lymphocyte counts; RA: room air; LDH: Lactate dehydrogenase; SD: standard deviation; IQR: interquartile range.

respectively, $p = 0.022$) and post-bronchodilator (84.0% vs 98.4% and 100.3% respectively, $p = 0.013$). The values of FEV1/FVC in the non-severe pneumonia and the severe pneumonia group were much less than that in the mild symptom group, with statistical significance in both pre bronchodilator (82.6% and 82.9% vs 86.6% respectively, $p = 0.019$) and post bronchodilator (83.7% and 82.9% vs 87.8% respectively, $p = 0.007$). Pre bronchodilator FEF25-75% in the mild symptom, non-severe pneumonia and severe pneumonia groups were 100.3 ± 26.9%, 91.0 ± 23.8% and 85.3 ± 32.9%, respectively. Post bronchodilator FEF25-75% in the mild symptom, non-severe pneumonia and severe pneumonia groups were 106.4 ± 25.8%, 95.7 ± 25.1% and 86.9 ± 38.0%, respectively.

Several cases of abnormalities in spirometry were detected. At day 60 after onset of symptoms, 5 patients (5.7%) had an FEV1/FVC ratio of < 75%; 7 patients (8%) had an FVC < 80%; 8 patients (9.2%) had an FEV1 < 80%; 8 (9.2%) had an FEF25-75 < 65% and 3 (3.4%) had an PEF < 80% in the prebronchodilator test. Regarding the postbronchodilator test, 5 patients (5.7%) had an FEV1/FVC ratio of < 75%; 7 patients (8%) had an FVC < 80%; 5 (5.7%) had an FEV1 < 80%; 10 (11.5%) had an FEF25-75 <65% and 5 (5.7%) had an PEF <80%.

Overall abnormalities in spirometry were seen in 15 cases (17.2%). In the severe pneumonia group, there were 71.4% having abnormal spirometry. Meanwhile, abnormal spirometry was found at 10.2% and 15.6% in the mild symptom and non-severe pneumonia groups,

**Table 2. Results of pulmonary function tests, 6MWT and chest x-ray.**

| Parameters | Total (N = 87) | Mild symptom (N = 45) | Non-severe pneumonia (N = 35) | Severe pneumonia (N = 7) | P-value |
|---|---|---|---|---|---|
| **Spirometry*** | | | | | |
| FVC (% of predicted), mean ± SD | | | | | |
| *prebronchodilator* | 97.9 ± 14.1 | 98.1 ± 14.0 | 100.4 ± 13.8 | 84.4 ± 10.3 | 0.022 |
| *postbronchodilator* | 98.0 ± 13.7 | 98.4 ± 13.6 | 100.3 ± 12.9 | 84.0 ± 10.9 | 0.013 |
| FEV1 (% of predicted), mean ± SD | | | | | |
| *prebronchodilator* | 98.3 ± 13.5 | 99.7 ± 12.4 | 98.8 ± 14.7 | 86.7 ± 10.0 | 0.058 |
| *postbronchodilator* | 99.5 ± 13.4 | 101.4 ± 12.5 | 99.7 ± 13.8 | 86.3 ± 10.8 | 0.019 |
| FEV1/FVC (%), mean ± SD | | | | | |
| *prebronchodilator* | 84.7 ± 6.6 | 86.6 ± 6.8 | 82.6 ± 5.1 | 82.9 ± 8.2 | 0.019 |
| *postbronchodilator* | 85.8 ± 6.4 | 87.8 ± 6.6 | 83.7 ± 5.1 | 82.9 ± 7.4 | 0.007 |
| FEF25-75 (% of predicted), mean ± SD | | | | | |
| *prebronchodilator* | 95.4 ± 26.4 | 100.3 ± 26.9 | 91.0 ± 23.8 | 85.3 ± 32.9 | 0.168 |
| *postbronchodilator* | 100.5 ± 27.1 | 106.4 ± 25.8 | 95.7 ± 25.1 | 86.9 ± 38.0 | 0.080 |
| PEF (% of predicted), mean ± SD | | | | | |
| *prebronchodilator* | 107.1 ± 17.0 | 105.8 ± 17.8 | 110.3 ± 16.1 | 98.9 ± 14.5 | 0.206 |
| *postbronchodilator* | 109.3 ± 17.5 | 109.7 ± 15.1 | 111.7 ± 19.5 | 95.0 ± 17.3 | 0.068 |
| Spirometry Interpretation, number (%) | | | | | |
| *Abnormal spirometry* | 15 (17.2) | 5 (10.6) | 5 (15.2) | 5 (71.4) | 0.001 |
| *Restrictive defect* | 7 (8) | 3 (6.4) | 2 (6.1) | 2 (28.6) | 0.114 |
| *Obstructive defect and small airway disease¶* | 8 (9.2) | 2 (4.3) | 3 (9.1) | 3 (42.9) | 0.011 |
| **Six-minute walk distance (6MWD), meters, mean ± SD** | | | | | |
| 6MWD, mean ± SD | 529.9 ± 57.4 | 538.0 ± 56.8 | 527.5 ± 53.5 | 490.4 ± 70.8 | 0.118 |
| **Chest radiograph, number (%)** | | | | | |
| Residual lung fibrosis on CXR | 12 (13.8) | 0 | 6 (17.1) | 6 (85.7) | <0.001 |

* The results were expressed as a percentage of predicted values using normal values for the population of Thailand [17].

¶ Obstructive defect—mild symptom = 2 (4.4%), non-severe pneumonia = 2 (5.7%) and severe pneumonia = 1 (14.3%)

Small airway—mild symptom = 0, non-severe pneumonia = 1 (2.9%) and severe pneumonia = 2 (28.6%)

FVC: forced vital capacity; FEV: forced expiratory volume in the first second; FEF: forced expiratory flow; PEF: peak expiratory flow; 6MWD: six-minute walk distance; CXR: chest radiograph.

respectively. There were restrictive defects in 2 patients (28%) in the severe pneumonia group. There were also 3 patients (42.9%) in the severe pneumonia group having obstructive defects in which 2 out of the 3 were small airway disease. Regarding patients in the mild symptom and non-severe pneumonia groups, only 2 (4.3%) and 3 patients (9.1%), respectively, had obstructive defects (Table 2). Among the patients with obstructive defect and the small airway disease group, six patients (75%) had a history of previous smoking, one (12.5%) was a current smoker and the other was a non-smoker.

## 6MWT

The mean 6-min walking distance (6MWD) in all subjects was 529.9 ± 57.4 meters. The mild symptom and non-severe pneumonia groups had 6MWDs means of 538 ± 56.8 and 527.5 ± 53.5 meters, respectively. The severe pneumonia group tended to have a shorter 6MWD mean, 490.4 ±70.8 meters, but was not statistically significant (*p* = 0.118). The mean oxygen saturation pre 6MWT in the mild symptom, non-severe pneumonia and severe

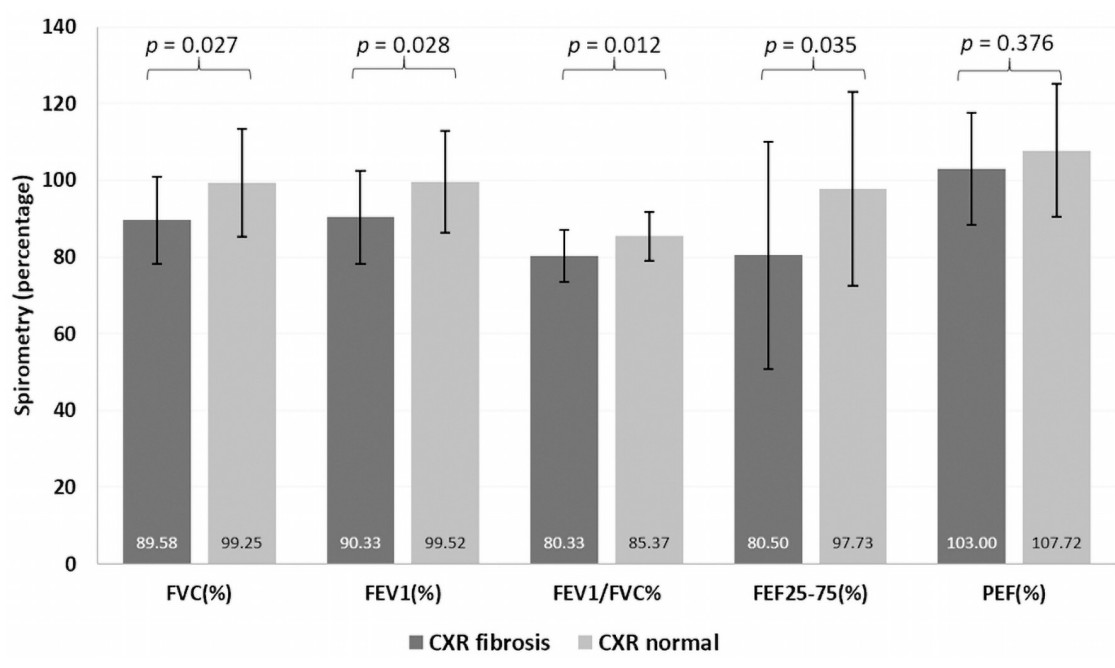

**Fig 2. Correlation between residual lung fibrosis on chest radiograph and spirometry.**

pneumonia groups were 98.11%, 98.03% and 97.86%, respectively ($p = 0.846$). The mean oxygen saturation post 6MWT in the corresponding groups were 98.0%, 97.89% and 97.14%, respectively ($p = 0.201$). There were no differences in oxygen saturation among groups for both pre and post 6MWT.

## Correlation between chest radiographs with lung function and 6MWD

At follow-up visit of 60 days after onset of symptoms, 12 patients (13.8%) had abnormal chest radiographs that showed residual fibrosis. In the severe pneumonia group, 6 of 7 patients (85.7%) had residual pulmonary fibrosis while this abnormal chest radiograph was found at only 7.5% in others ($p<0.001$).

The mean FVC was significantly lower in patients with abnormal chest radiographs when compared to those with normal chest radiographs, both at prebronchodilator (89.6±11.4 vs 99.3±14.1, $p = 0.027$) and postbronchodilator periods (89.4±11.5 vs 99.4±14.1 $p = 0.018$ in). Other parameters such as $FEV_1$, $FEV_1$/FVC and FEF25-75% were also significantly lower in patients who had residual lung fibrosis on chest radiographs when compared to patients with a normal chest radiograph (Fig 2).

The mean 6MWD was lower in patients with abnormal chest radiographs when compared to those with a normal chest radiograph, but this difference was not statistically significant (518.5±78.3 vs 531.8± 53.8, $p = 0.461$).

## Discussion

The results from the present study have shown that after the patients recovered from COVID-19, 60 days counted from the onset of symptoms, the average values of pulmonary function were within the normal range. However, when we analyzed the results of pulmonary function

testing for individual patients, there were a total of 15 patients (17.2%) with pulmonary function impairments. These included 7 patients (8%) with restrictive defect and 8 patients (9.2%) with obstructive defect. The number of abnormal pulmonary functions in the present study was found to be less than that from previous studies [9, 10, 12]. This could be explained mostly by two reasons. First, our study included all patients with laboratory-confirmed diagnosis, ranging from mild symptoms to non-severe and severe pneumonia. Second, total lung capacity (TLC) and diffusing capacity for carbon monoxide (DLCO) were not performed in this study. DLCO is the most common abnormal parameters in both COVID-19 and SARS patients. Therefore, only FVC, FEV1/FVC, FEF25-75% abnormalities were observed in this study.

In the previous studies [10, 12, 13], there were no significant differences in FVC and FEV1/FVC between the non-severe and severe groups. However, the results from the present study were significantly different from the previous studies that FVC was lower in the severe pneumonia group, than in the other groups. This is because 85.7% of the severe pneumonia patients were found to have abnormal CXR with residual lung fibrosis. Thus, patients in this group could have a low FVC that represented as a restrictive defect.

Another difference compared to the previous studies, the $FEV_1/FVC$ tended to be lower in the severe pneumonia group. This suggests that the severe pneumonia group had more obstructive defect than the others. This could be explained by two reasons. First, it may be due to undiagnosed preexisting airway disorders in patients with severe pneumonia. Interestingly, it was found that in patients with obstructive defect, the history of smoking was 87.5%, with 75% former smokers and 12.5% current smokers. The second reason is that COVID-19 disease can lead to abnormalities in the bronchi. An autopsy of deceased COVID-19 patients showed necrotizing bronchiolitis [6] and focal bronchial or bronchiolar inflammation [5], which could explain the small airway dysfunction. Therefore, long-term monitoring of chest symptoms and spirometry should be considered in patients recovered from COVID-19, in order to determine progression of the obstructive lung defect.

The 6MWT results showed no statistically significant difference among groups, but it was found to be substantially lower in the severe pneumonia group. The results were concordant to that from a previous study [10]. Nevertheless, the limitation is the factors interfere the 6MWT interpretation. According to the guidelines for the six-minute-walk test from ATS statement [18], the older age and higher weight can reduce the six-minute walk distance. Likewise, in our study, patient in the severe pneumonia group are older and heavier than the others.

A total of 85.7% in the severe pneumonia group had residual lung fibrosis on chest radiographs and those with lung fibrosis also had abnormal lung function. Therefore, patients recovered from severe pneumonia should be closely monitored for symptoms, chest radiographs, pulmonary function, and may be considered for additional pulmonary rehabilitation.

The study limitation is that lung volume and DLCO were not tested. In the study of patients infected with SARS, the most common pulmonary function abnormalities were reduced DLCO, followed by TLC and FVC [7, 19]. It was also found in a previous study in China that the most common abnormality in COVID-19 patients was DLCO. In this study, TLC and DLCO were not performed. Therefore, the results of this study only demonstrated abnormalities of FVC $FEV_1$, $FEV_1$/ FVC FEF25-75 and PEF. Another limitation is the absence of baseline spirometry, making it difficult to distinguish whether the obstructive defect arose from the underlying airway disease or due to COVID-19 infection. The amount of the patient is also the limitation of this study as the first pandemic wave of Thailand ended with the total case of less than 4,000. The total cases admitted at our hospital was only 147, we intend to follow up all of them but some of them went back to hometown and could not come to follow up. However,

the population of the missing group were not different from the study group in term of severity.

## Conclusions

Severe COVID-19 pneumonia patients had a higher rate of abnormal spirometry and had more frequent residual pulmonary fibrosis on a chest radiograph when compared to patients with mild symptoms and non-severe pneumonia. Long-term following up of patients after recovery from SAR-CoV-2 infection is essential, especially in severe pneumonia groups.

## Supporting information

**S1 Table. Laboratory results of 87 recovered COVID-19 patients during follow up period (60 day after onset of symptoms).**
(DOCX)

**S2 Table. Spirometry results of 87 recovered COVID-19 patients.**
(DOCX)

## Acknowledgments

We are grateful to all study patients and all attending staffs, fellows, residents and nurses of Chakri Naruebodindra Medical Institute, Faculty of Medicine Ramathibodi Hospital, Mahidol University.

## Author Contributions

**Conceptualization:** Dararat Eksombatchai, Thananya Wongsinin, Somnuek Sungkanuparph.

**Data curation:** Dararat Eksombatchai, Thananya Wongsinin, Thanyakamol Phongnarudech.

**Formal analysis:** Dararat Eksombatchai, Somnuek Sungkanuparph.

**Investigation:** Dararat Eksombatchai, Thananya Wongsinin, Thanyakamol Phongnarudech.

**Methodology:** Dararat Eksombatchai, Thananya Wongsinin, Somnuek Sungkanuparph.

**Supervision:** Somnuek Sungkanuparph.

**Visualization:** Dararat Eksombatchai, Thananya Wongsinin, Somnuek Sungkanuparph.

**Writing – original draft:** Dararat Eksombatchai, Thananya Wongsinin, Thanyakamol Phongnarudech, Kanin Thammavaranucupt, Naparat Amornputtisathaporn, Somnuek Sungkanuparph.

**Writing – review & editing:** Dararat Eksombatchai, Thananya Wongsinin, Thanyakamol Phongnarudech, Kanin Thammavaranucupt, Naparat Amornputtisathaporn, Somnuek Sungkanuparph.

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
