## [Decision Letter · Decision Letter 0]

27 May 2021

PONE-D-21-11081

Pulmonary function and six-minute-walk test in patients after recovery from COVID-19: A prospective cohort study

PLOS ONE

Dear Dr. Wongsinin,

Thank you for submitting your manuscript to PLOS ONE. After careful consideration, we feel that it has merit but does not fully meet PLOS ONE’s publication criteria as it currently stands. Therefore, we invite you to submit a revised version of the manuscript that addresses the points raised during the review process.

We look forward to receiving your revised manuscript.

Kind regards,

Tai-Heng Chen, M.D.

Academic Editor

PLOS ONE

2. During the internal evaluation of the manuscript we have noted a discrepancy in the informed consent procedure between the ethics statement in the online submission form and the manuscript text. Please provide some clarification regarding this matter.

3. Furthermore, please provide additional information regarding the source of the demographic information presented in table 1.

Reviewers' comments:

Reviewer's Responses to Questions

**Comments to the Author**

1. Is the manuscript technically sound, and do the data support the conclusions?

Reviewer #2: Yes

Reviewer #3: Partly

Reviewer #5: No

2. Has the statistical analysis been performed appropriately and rigorously? 

Reviewer #2: No

Reviewer #3: No

Reviewer #5: No

3. Have the authors made all data underlying the findings in their manuscript fully available?

Reviewer #2: Yes

Reviewer #3: Yes

Reviewer #5: No

4. Is the manuscript presented in an intelligible fashion and written in standard English?

Reviewer #2: Yes

Reviewer #3: Yes

Reviewer #5: Yes

5. Review Comments to the Author

Reviewer #2: This research content is very interesting. However, there remain several concerns to be clarified, some of which are critical.

Concerns or questions:

1. This study is described as a prospective cohort study, which is questionable because time factors were not taken into account. Isn't it a case-control study?

2. We have data 60 days after hospital discharge, but no data on background factors at the time of hospital discharge? In the first place, the background factors of the three groups are different, so it seems impossible to say what influenced the results.

3. The flow of research subjects should be described, although it seems to have been decided on 87 cases because of subjects, deficiencies, and omissions.

4. You are comparing means, but is the data you are using normally distributed as well (Table 1, 2)?

5. The target population is relatively young, but in my country, mature to elderly people are more likely to suffer from diseases, why young? How did you decide who to include? Who was excluded? Any deficiencies or omissions?

6. The diagram is hard to see, please fix it (Figure.1).

7. In line 200, Figure.1, D-dimer and LDH are mentioned, but is it necessary to show this data in this study? Does it fit the purpose of this study?

8. Lines 105-106, "Pneumonia was defined as clinical symptoms of respiratory tract infection with abnormal lung imaging compatible with COVID-19 pneumonia” is a subjective assessment, but is it accurate? It seems to lack objectivity. Is it possible to sufficiently classify the subjects?

9. The limitations of the study should be stated differently because of the effects of confounding factors and many other things.

Reviewer #3: Eksombatchai et al. report the pulmonary function after the patients recovered from COVID-19. Specifically, the authors found that higher prevalence rates of abnormal spirometry and residual fibrosis on the chest radiographs in the severe symptom group. This paper may be an interesting paper, especially in pulmonary rehabilitation. However, I have some major concerns, which interrupt the acceptance of the paper in its present form.

Major issues: 1

Although the study and results look to be reported properly, I do not feel that the statistical description fully meets. For example, the authors should describe F-value（line 145-146, 153）. And in fig1 and fig2, I could not find out what kind of the statistical analysis the authors use in two groups comparison.

Major issues: 2

Although the results of Age and BMI were statistically significant in the paper, that are poorly discussed in the Discussion section. For example, in line 165-167, is not the 6MWT affected these factors?

Major issues: 3

In Line 70-71, 74, 106-109, 118-119, the authors must cite the proper references. The authors must refer to "Submission Guidelines”.

Minor issues: 1

There are no error bars in the figures.

Minor issues: 2

In the table1, "*" indicate "follow up" and "p-value".

Minor issues: 3

Please report the overlapping of diseases（Line 140-142）.

Minor issues: 4

Please revise the abbreviations in the manuscript.

In line 36, "MWD" appeared the first time. "WHO" is abbreviated twice（Line 72, 138-139）. Is six-minute-walk distance "6MWD”? Is six-minute-walk test "6MWT”? （Line 55）

Reviewer #5: Peer Review for PONE-D-21-11081

“Pulmonary function and six-minute-walk test in patients after recovery from COVID-19: A prospective cohort study”

1. Summary of the research and your overall impression

Dararat Eksombatchai et al. reported spirometry abnormalities in 87 patients recovering from COVID-19; at follow-up on day 60 after onset of symptoms. They observed spirometry abnormalities in 15 cases (17.2%), most of which could be attributed to the patient group with severe pneumonia. They described that the group with severe pneumonia had a shorter mean 6-minute walk distance and more frequent abnormal chest radiographs. The research topic is interesting with high relevance to clinical practice and pulmonary rehabilitation.

I think that this work could be an important addition to the field, but there are some major problems, especially with the design and data of the study, that cannot be overlooked. Therefore, I do not recommend this work for publication.

I. Essential points the authors need to address before such work can proceed, are:

1) Biggest issue is this study does not have a sufficiently large sample size to answer the research question of interest. In particular, the group with severe pneumonia (n=7) is too small to draw general conclusions. The comparison groups are younger - better matching would therefore be desirable.

Title of paper, experimental design, data, and conclusions are inconsistent and do not meet high medical research standards. Study design and implementation are difficult to follow, the selection of the outcomes appears selective and not well justified ... see my comments below:

2) Line 89-92 and 96 f.: The description of the sample and conditions is insufficient. How many COV-patients were treated (acutely, t1) at the study center, with which severity (inpatient, outpatient, ICU, ventilation, ...). How long was the acute treatment/inpatient stay? ... How many of them came for follow-up after 60 days (t2, drop-out?; selection bias?; cf. point 10).

3) Line 110: It is a pity that results of measurements e.g., pulmonary function testing is only available for the follow-up period. This weakens the study design and the significance of the results.

II. Still important minor issues, but not affecting the overall conclusions of the manuscript are:

4) Ethics Statement: “Consent was not obtained because the data were analyzed anonymously.”?

5) Data Availability: “Yes - all data are fully available without restriction” – where? …

“All relevant data are within the manuscript and its Supporting Information files.” – is not given

6) Line 27: Please specify the observation period; data on inpatient length of stay are missing

7) Line 40: If a difference is not significant, it cannot be longer or shorter – this should be formulated differently (it makes clear, among other things, that the group of severe pneumonia is too small).

8) Line 88: How does it compare to pneumonia not caused by COV19?

9) Line 98: The study does not give the appearance of being prospective in its presentation …

10) Line 105: … not clear to the reader … The classification of severity was done at the follow-up time?!!? … information missing, how was this exactly measured

11) Line 111: cf. point 4 - Ethics Statement!! … not the same

12) Line 124: the choice of method would have argued for a MANOVA with repeated measures*, how was alpha error accounted for, what would a sample estimate with the observed effect sizes yield ...

*eg. Line 133: Laboratory results baseline – peak? – follow up; the selection of the outcomes appears selective, but is not well justified …

13) Line 144: The different group characteristics are not well described, e.g. the age difference between the groups is not mentioned

14) Line 152: cf. point 7

15) Line 156: It would be important to have a comparison to disease onset/t1 for this purpose

16) Line 198 vs. line 175: several cases of abnormalities in 71.4% (= 5 out of 7) having abnormal spirometry … Indicates a large heterogeneity and thus an insufficient amount of data and control of possible confounders

17) Table 2: In addition to t1 data, the raw values would also be of interest here. Unfortunately, the related publication [13] with the standard values is not publicly available – … and should be mentioned here once again...

18) Line 200 and Figures: Grouping (obstructive/restrictive) and description is difficult to follow ... a more detailed description of these two groups (n!!, age ..) is missing?

19) Line 219: how was this determined …

It is not helpful is recommending further data collection. I hope the authors continue their research in this area, which looks promising and has important implications.... but in my opinion, this exploratory study documentation is not yet ready for scientific publication in a high-impact journal.

6. PLOS authors have the option to publish the peer review history of their article (what does this mean?). If published, this will include your full peer review and any attached files.

Reviewer #2: No

Reviewer #3: No

Reviewer #5: No

---

## [Author Response · Author response to Decision Letter 0]

26 Jul 2021

Response to Reviewers

Response: Thank you for the suggestion on our manuscript. We have considered all the comments in each point and put our effort to respond the requested revisions. We believe that our reformed manuscript is able to clarify all the uncertain issues.

1. Is the manuscript technically sound, and do the data support the conclusions?

Reviewer #2: Yes

Reviewer #3: Partly

Reviewer #5: No

Response: Thank you for the comments. We have organized the data to support the conclusions in our manuscript.

2. Has the statistical analysis been performed appropriately and rigorously?

Reviewer #2: No

Reviewer #3: No

Reviewer #5: No

Response: Thank you. We add the detail of the analyzing method and revise some improper statistical form as below.

3. Have the authors made all data underlying the findings in their manuscript fully available?

Reviewer #2: Yes

Reviewer #3: Yes

Reviewer #5: No

Response: We have added the essential data to fulfill our manuscript.

4. Is the manuscript presented in an intelligible fashion and written in standard English?

Reviewer #2: Yes

Reviewer #3: Yes

Reviewer #5: Yes

Response: Thank you

5. Review Comments to the Author

Reviewer #2: This research content is very interesting. However, there remain several concerns to be clarified, some of which are critical.

Concerns or questions:

1. This study is described as a prospective cohort study, which is questionable because time factors were not taken into account. Isn't it a case-control study?

Response: To clarify the study design, we add the flow of the recruited patients (Figure1) in our manuscript. From the beginning, we planed and designed the study to follow up pulmonary function test in all the admitted patients after discharging from the hospital and at 60 days after the first date of the symptom. 

2. We have data 60 days after hospital discharge, but no data on background factors at the time of hospital discharge? In the first place, the background factors of the three groups are different, so it seems impossible to say what influenced the results. 

Response: According to our purpose, we intend to evaluate the lung function after recovery by categorized the patient into three groups, which classify based on clinical course of patient while admitting in our hospital. And we discharge the patient on the basis of good clinical outcome including oxygenation, and laboratories. The baseline factors that were different among groups i.e. age, underlying diseases, were correlated with the severity, but not directly influenced the primary outcomes. 

3. The flow of research subjects should be described, although it seems to have been decided on 87 cases because of subjects, deficiencies, and omissions.

Response: To make it clear, we have added the flow of recruited patients in our manuscript. 

Line 139 --- “147 Patients were diagnosed and admitted with COVID-19 in Chakri Naruebodindra Medical Institute, Faculty of Medicine Ramathibodi Hospital during the study period. Of all, two patients died due to severe pneumonia and multiorgan failure; 52 patients did not participate in this study secondary to inconvenient long-distance transportation from their hometown; and six patients failed to finish the spirometry test.”

Line 145 --- Flow of the research subjects (Fig 1)

4. You are comparing means, but is the data you are using normally distributed as well (Table 1, 2)?

Response: We use Shapiro-wilk to test for normal distribution. We found that D-dimer is not normally distributed. Therefore, this factors is presented as median and IQR, as in the table 1. The differences of this variable between groups are compared using Kruskal-Wallis. 

Line 124 --- “Non-normally distributed variables are shown as median (interquartile range [IQR]).”

Line 127 --- “The differences of non-normally distributed variable between groups are compared using Kruskal-Wallis.”

5. The target population is relatively young, but in my country, mature to elderly people are more likely to suffer from diseases, why young? How did you decide who to include? Who was excluded? Any deficiencies or omissions?

Response: According to the flow of research subjects, we included all the patients who admitted in our center during the study period, and excluded only the ones who died or decided not to participate due to inconvenient transportation from their hometown. In our center, we admitted all the patients who had PCR-confirmed COVID-19 regardless of the severity. In the first wave of the pandemic in 2020, there were 147 patients which the mean age was 39.6 � 11.8 year, admitted in our hospital. As a result of the immediate lock-down policy, overall case in Thailand were only 3,151 cases in the two months of the endemic period. Moreover, the spreading was limited in the places popular for the young people such as pub, concert, and boxing stadium. We have analyzed the age and gender between our study patients and those who decided not to participate in this study and found that there were no differences. 

6. The diagram is hard to see, please fix it (Figure.1).

Response: Due to the irrelevant information mentioned in the comment number 7, we decided to remove the diagram which is not the main purpose of the research. 

7. In line 200, Figure.1, D-dimer and LDH are mentioned, but is it necessary to show this data in this study? Does it fit the purpose of this study?

Response: It may not fit the purpose at first but from the analyzing data we found that the parameters may help us predict the pulmonary function outcome of COVID-19, we illustrated in the document. However, after carefully consideration, we decide to remove it from our manuscript to remain only the data focus on our main objective. 

8. Lines 105-106, "Pneumonia was defined as clinical symptoms of respiratory tract infection with abnormal lung imaging compatible with COVID-19 pneumonia” is a subjective assessment, but is it accurate? It seems to lack objectivity. Is it possible to sufficiently classify the subjects?

Response: According to WHO “COVID-19 pneumonia” definition, the pneumonia is diagnosed by the clinical sign of pneumonia and chest imaging. We classified the pneumonia patient from their positive clinical signs, and lung imaging which confirmed by chest radiologist in co-operate with pulmonologists in our center.

9. The limitations of the study should be stated differently because of the effects of confounding factors and many other things.

Response: Thank you for the suggestion about limitations. We stated the further limitations of our study in the discussion part, describe as follow:

1) Factors reducing six-minute-walk test (6MWT) are older age and higher body weight.

line 265 --- “Nevertheless, the limitation is the factors interfere the 6MWT interpretation. According to the guidelines for the six-minute-walk test from ATS statement, the older age and higher weight can reduce the six-minute walk distance. Likewise, in our study, patient in the severe pneumonia group are older and heavier than the others.”

2) The small number of the patients are limiting due to a low total national case and the follow up issue. 

Line 280 --- “The amount of the patient is also the limitation of this study as the first pandemic wave of Thailand ended with the total case of less than 4,000. The total cases admitted at our hospital was only 147, we intend to follow up all of them but some of them went back to hometown and could not come to follow up. However, the population of the missing group were not different from the study group in term of severity.”

Group Follow up plan (145 patients) Study population (87 patients)

Mild symptoms, n(%) 71 (49.0%) 45 (51.7%)

Non-severe pneumonia, n(%) 56 (38.6%) 35 (40.2%)

Severe pneumonia, n(%) 18 (12.4%) 7 (8.0%)

Reviewer #3: Eksombatchai et al. report the pulmonary function after the patients recovered from COVID-19. Specifically, the authors found that higher prevalence rates of abnormal spirometry and residual fibrosis on the chest radiographs in the severe symptom group. This paper may be an interesting paper, especially in pulmonary rehabilitation. However, I have some major concerns, which interrupt the acceptance of the paper in its present form.

Major issues: 1

Although the study and results look to be reported properly, I do not feel that the statistical description fully meets. For example, the authors should describe P-value line 145-146, 153）. And in fig1 and fig2, I could not find out what kind of the statistical analysis the authors use in two groups comparison.

Response: 

-We have revised the baseline characteristic in table 1 in order to make it easy to understand and correct some inaccurate data. We removed the data about laboratories at peak and follow up, remained only the data on admission. We also changed the oxygen saturation level from the follow up data to the data on admission which the P-value was revised to <0.001. 

Line 156 --- “The patients in the severe pneumonia group had a higher BMI and older age than those in the mild symptom and non-severe pneumonia groups. The mean oxygen saturation on admission was significantly lower in severe pneumonia group compared with the mild symptom and non-severe pneumonia groups. The severe pneumonia group had higher levels of serum lactate dehydrogenase (LDH) compared with the mild symptom and non-severe pneumonia groups. This finding was similar to the levels of serum D-dimer that was highest among severe cases. Meanwhile, there were no significant differences in serum creatinine between the three groups (p=0.282). The mean absolute lymphocyte count was lower in the severe pneumonia group during the admission. The average length of stay was significantly higher in severe pneumonia group compared with non-severe pneumonia, and mild symptom group (24.9±5.6 vs 12.9±5.8 and 8.0±2.5 respectively, p<0.001).”

-In figure 1 and 2, we use independent sample T-test. 

Line 128 ---- “Independent sample T-test was used to figure out the correlation between chest radiography with lung function and 6MWD.”

Major issues: 2

Although the results of Age and BMI were statistically significant in the paper, that are poorly discussed in the Discussion section. For example, in line 165-167, is not the 6MWT affected these factors?

Response: Thank you. We have added the discussion about this issue.

1) Factors reducing six-minute-walk test (6MWT) are older age and higher body weight.

Line 265 --- “Nevertheless, the limitation is the factors interfere the 6MWT interpretation. According to the guidelines for the six-minute-walk test from ATS statement, the older age and higher weight can reduce the six-minute walk distance. Likewise, in our study, patient in the severe pneumonia group are older and heavier than the others.”

 2) BMI vs spirometry

The patients in the severe pneumonia group had a higher BMI than those in the mild symptom and non-severe pneumonia groups. This factor may cause more restrictive defect from obesity in severe pneumonia group than others. However, there was no difference in BMI in restrictive group compared to normal spirometry group (24.6±5.7 vs 23.1±4.0). 

Major issues: 3

In Line 70-71, 74, 106-109, 118-119, the authors must cite the proper references. The authors must refer to "Submission Guidelines”.

Response: Thank you for the suggestion. We have revised references according to submission guideline.

Line 70-71 

“Zhu N, Zhang DY, Wang WL, et al. A novel coronavirus from patients with pneumonia in China, 2019. N Engl J Med. 2020 doi: 10.1056/nejmoa2001017.”

Line 74

“World Health Organization. COVID-19 Weekly Epidemiological Update. Mar 30, 2021. [cited 2021/3/30]; Available from https://www.who.int/emergencies/diseases/novel-coronavirus-2019/situation-reports.”

Line 106-109

“World Health Organization. Clinical management of severe acute respiratory infection when Novel coronavirus (nCoV) infection is suspected: interim guidance. January 28, 2020. [cited 2020/4/15]; Available from: https://www.who.int/publications-detail/clinical-management-of-severe-acute-respiratoryinfection-when-novel-coronavirus-(ncov)-infection-is-suspected. ”

Line 118-119

“Graham BL, Steenbruggen I, Miller MR, Barjaktarevic IZ, Cooper BG, Hall GL, et al. Standardization of Spirometry 2019 Update. An Official American Thoracic Society and European Respiratory Society Technical Statement. Am J Respir Crit Care Med. 2019 Oct 15;200(8):e70–88.”

Minor issues: 1

There are no error bars in the figures.

Response: We have added error bar in the figure 2.

Minor issues: 2

In the table1, "*" indicate "follow up" and "p-value". Response: We have removed "*" from table1.

Minor issues: 3

Please report the overlapping of disease（Line 140-142）.

Response: We add overlapping disease in line 150.

“Among the patient with the underlying diseases, there were some overlapping of them. The number of the patients with DM and HT, HT and DLP, and DM, HT and DLP were 2 (2.3%), 3 (3.4%), and 1 (1.1%), respectively.”

Minor issues: 4

Please revise the abbreviations in the manuscript.

In line 36, "MWD" appeared the first time. "WHO" is abbreviated twice（Line 72, 138-139）. Is six-minute-walk distance "6MWD”? Is six-minute-walk test "6MWT”? （Line 55） Response: We have revised the abbreviations in the manuscript.

Reviewer #5: Peer Review for PONE-D-21-11081

“Pulmonary function and six-minute-walk test in patients after recovery from COVID-19: A prospective cohort study”

1. Summary of the research and your overall impression

Dararat Eksombatchai et al. reported spirometry abnormalities in 87 patients recovering from COVID-19; at follow-up on day 60 after onset of symptoms. They observed spirometry abnormalities in 15 cases (17.2%), most of which could be attributed to the patient group with severe pneumonia. They described that the group with severe pneumonia had a shorter mean 6-minute walk distance and more frequent abnormal chest radiographs. The research topic is interesting with high relevance to clinical practice and pulmonary rehabilitation.

I think that this work could be an important addition to the field, but there are some major problems, especially with the design and data of the study, that cannot be overlooked. Therefore, I do not recommend this work for publication.

I. Essential points the authors need to address before such work can proceed, are:

1) Biggest issue is this study does not have a sufficiently large sample size to answer the research question of interest. In particular, the group with severe pneumonia (n=7) is too small to draw general conclusions. The comparison groups are younger - better matching would therefore be desirable.

Title of paper, experimental design, data, and conclusions are inconsistent and do not meet high medical research standards. Study design and implementation are difficult to follow, the selection of the outcomes appears selective and not well justified ... see my comments below:

Response: Thank you for the suggestion.

--this study does not have a sufficiently large sample size to answer the research question of interest.

Response: The number of the patients are limited due to a low total national cases and the follow up issue. However, our cohort may represent the whole picture of natural course of COVID-19 since we have admitted all patients with COVID-19 and included all variety of disease severity. In addition, the results from our study are be able to show the significant abnormal spirometry. 

Add line 280 --- “The amount of the patient is also the limitation of this study as the first pandemic wave of Thailand ended with the total case of less than 4,000. The total cases admitted at our hospital was only 147, we intend to follow up all of them but some of them went back to hometown and could not come to follow up. However, the population of the missing group were not different from the study group in term of severity.”

Group Follow up plan (145 patients) Study population (87 patients)

Mild symptoms, n(%) 71 (49.0%) 45 (51.7%)

Non-severe pneumonia, n(%) 56 (38.6%) 35 (40.2%)

Severe pneumonia, n(%) 18 (12.4%) 7 (8.0%)

--The comparison groups are younger - better matching would therefore be desirable.

Response: We regarded the difference of the age between groups so we used the age-adjusted reference values of the spirometry which is the main outcome. (Dejsomritrutai W, Nana A, Maranetra KN, Chuaychoo B, Maneechotesuwan K, Wongsurakiat P, et al. Reference spirometric values for healthy lifetime nonsmokers in Thailand. J Med Assoc Thai 2000; 83(5): 457-466.)

--Title of paper, experimental design, data, and conclusions are inconsistent and do not meet high medical research standards. Study design and implementation are difficult to follow, the selection of the outcomes appears selective and not well justified 

Response: We have added the flow of the research subjects (Figure 1) and removed some irrelevant data to concise the manuscript (line 200). We believe this revised manuscript is easier to understand and precisely answers our main objective. 

2) Line 89-92 and 96 f.: The description of the sample and conditions is insufficient. How many COV-patients were treated (acutely, t1) at the study center, with which severity (inpatient, outpatient, ICU, ventilation, ...). How long was the acute treatment/inpatient stay? ... How many of them came for follow-up after 60 days (t2, drop-out?; selection bias?; cf. point 10).

Response: we have added the flow of recruited patients in our manuscript. 

Line 139 --- “147 Patients were diagnosed and admitted with COVID-19 in Chakri Naruebodindra Medical Institute, Faculty of Medicine Ramathibodi Hospital during the study period. Of all, two patients died due to severe pneumonia and multiorgan failure; 52 patients did not participate in this study secondary to inconvenient long-distance transportation from their hometown; and six patients failed to finish the spirometry test.”

Line 145 --- Flow of the research subjects (Figure 1)

According to the flow of research subjects, we included all the patients who admitted in our center during the study period, and excluded only the ones who died or decided not to participate due to inconvenient transportation from their hometown. In our center, we admitted all the patients who had PCR-confirmed COVID-19 regardless of the severity. In the first wave of the pandemic in 2020, there were 147 patients which the mean age was 39.6 � 11.8 year, admitted in our hospital. As a result of the immediate lock-down policy, overall cases in Thailand were only 3,151 cases in the two months of the endemic period. Moreover, the spreading was limited in the places popular for the young people such as pub, concert, and boxing stadium. We have analyzed the age and gender between our study patients and those who decided not to participate in this study and found that there were no differences. 

3) Line 110: It is a pity that results of measurements e.g., pulmonary function testing is only available for the follow-up period. This weakens the study design and the significance of the results.

Response: It is a limitation of our study that we did not have a comparison data of pulmonary function test because of the limited resources and hospital infection control measures for COVID-19 patients that spirometry during admission is contagious and therefore prohibited.

II. Still important minor issues, but not affecting the overall conclusions of the manuscript are:

4) Ethics Statement: “Consent was not obtained because the data were analyzed anonymously.”?

Response: Thank you for the notice, it is our mistake on the manuscript submission process. Actually, we literally got the informed consent prior to the enrollment of all study patients.

5) Data Availability: “Yes - all data are fully available without restriction” – where? …

“All relevant data are within the manuscript and its Supporting Information files.” – is not given 

Response: We have added the statement “All relevant data are within the manuscript and its Supporting Information files” in the manuscript.

6) Line 27: Please specify the observation period; data on inpatient length of stay are missing

Response: We additionally put the length of stay in table 1.

7) Line 40: If a difference is not significant, it cannot be longer or shorter – this should be formulated differently (it makes clear, among other things, that the group of severe pneumonia is too small).

Response: Thank you for your suggestion. We have already adjusted our manuscript in order not to mislead the reader.

Line 39-41 --- “Although the severe pneumonia group tended to have a shorter mean 6-min walking distance, but this was not statistically significant (p=0.118).”

8) Line 88: How does it compare to pneumonia not caused by COV19?

Response: In COVID-19 pneumonia, there is a bronchiolitis feature in histopathology which may cause the obstructive pattern more than the others. (Bradley BT, Maioli H, Johnston R, Chaudhry I, Fink SL, Xu H, et al. Histopathology and ultrastructural findings of fatal COVID-19 infections in Washington State: a case series. The Lancet 2020; 396(10247): 320-332.) However, in line 88, we mentioned about the COVID-19 pneumonia.

9) Line 98: The study does not give the appearance of being prospective in its presentation …

Response: To clarify the study design, we added the flow of the recruited patients (Figure1) in our manuscript. From the beginning, we planed and designed the study to follow up pulmonary function test in all the admitted patients at 60 days after the first date of the symptom. 

10) Line 105: … not clear to the reader … The classification of severity was done at the follow-up time?!!? … information missing, how was this exactly measured

Response: Our aim of the study was to compare the recovery of the lung function in each severity category which was classified at the disease developing period. By the discharge time, all of our patients were improved including chest imaging, oxygenation, and clinical status. None of them needed the home oxygen therapy.

Line 109 Add “The classification of severity was done during the admission.” 

11) Line 111: cf. point 4 - Ethics Statement!! … not the same

Response: Thank you for the notice, it is our mistake on the submission process. Actually, we literally got the informed consent prior to the enrollment of all study patients.

12) Line 124: the choice of method would have argued for a MANOVA with repeated measures*, how was alpha error accounted for, what would a sample estimate with the observed effect sizes yield ...

*eg. Line 133: Laboratory results baseline – peak? – follow up; the selection of the outcomes appears selective, but is not well justified …

Response: This study collected the data from all our patients, not from the calculated number. And we did not focus on the time effect instead we tested the difference between the groups at the same time point (60 days after onset of symptoms). For the laboratory results, we removed the data about laboratories at peak and follow up, remained only the data on admission.

13) Line 144: The different group characteristics are not well described, e.g. the age difference between the groups is not mentioned

Response: Add age difference between group in line 156.

Line 156 --- “The patients in the severe pneumonia group had a higher BMI and older age than those in the mild symptom and non-severe pneumonia groups.”

14) Line 152: cf. point 7

Response: We have already adjusted our manuscript in order not to mislead the reader.

-We removed the data in line 152. We have revised the baseline characteristic in table 1 in order to make it easy to understand and correct some inaccurate data. We removed the data about laboratories at peak and follow up, remained only the data on admission.

15) Line 156: It would be important to have a comparison to disease onset/t1 for this purpose 

Response: We follow up spirometry at day 60 after onset of the symptoms for all patients because it is the safe timing to do spirometry. Due to the hospital infection control measures for COVID-19 patients, spirometry during the admission is contagious and prohibited.

16) Line 198 vs. line 175: several cases of abnormalities in 71.4% (= 5 out of 7) having abnormal spirometry … Indicates a large heterogeneity and thus an insufficient amount of data and control of possible confounders 

Response: Thank you for the comment. This is a limitation of our study and we have already mentioned in the discussion part.

Line 280 --- “The amount of the patient is also the limitation of this study as the first pandemic wave of Thailand ended with the total case of less than 4,000. The total cases admitted at our hospital was only 147, we intend to follow up all of them but some of them went back to hometown and could not come to follow up. However, the population of the missing group were not different from the study group in term of severity.”

17) Table 2: In addition to t1 data, the raw values would also be of interest here. Unfortunately, the related publication [13] with the standard values is not publicly available – … and should be mentioned here once again...

Response: -We have added reference of the standard values in table 2. 

-We attach the PDF file of reference spirometric values for healthy lifetime nonsmokers in Thailand. J Med Assoc Thai 2000; 83(5): 457-466. 

18) Line 200 and Figures: Grouping (obstructive/restrictive) and description is difficult to follow ... a more detailed description of these two groups (n!!, age ..) is missing?

Response: After carefully consideration, we decide to remove line 200 and figure1 from our research to focus at our main objective. 

19) Line 219: how was this determined …

Response: Abnormal chest radiographs that showed residual fibrosis were defined by there are residual reticular opacity compare with imaging during discharge time. Chest radiographs were confirmed by chest radiologist in co-operate with pulmonologists in our center.

It is not helpful is recommending further data collection. I hope the authors continue their research in this area, which looks promising and has important implications.... but in my opinion, this exploratory study documentation is not yet ready for scientific publication in a high-impact journal.

Response: thank you for reviewing our manuscript and give the very interesting suggestions and also indicate the crucial point for our study. We have concerned every point and try to organized the data in order to improve the manuscript.

---

## [Decision Letter · Decision Letter 1]

23 Aug 2021

Pulmonary function and six-minute-walk test in patients after recovery from COVID-19: A prospective cohort study

PONE-D-21-11081R1

Dear Dr. Wongsinin,

We’re pleased to inform you that your manuscript has been judged scientifically suitable for publication and will be formally accepted for publication once it meets all outstanding technical requirements.

Kind regards,

Tai-Heng Chen, M.D.

Academic Editor

PLOS ONE

Reviewers' comments:

Reviewer's Responses to Questions

**Comments to the Author**

1. If the authors have adequately addressed your comments raised in a previous round of review and you feel that this manuscript is now acceptable for publication, you may indicate that here to bypass the “Comments to the Author” section, enter your conflict of interest statement in the “Confidential to Editor” section, and submit your "Accept" recommendation.

Reviewer #2: All comments have been addressed

Reviewer #5: All comments have been addressed

2. Is the manuscript technically sound, and do the data support the conclusions?

Reviewer #2: (No Response)

Reviewer #5: Partly

3. Has the statistical analysis been performed appropriately and rigorously? 

Reviewer #2: (No Response)

Reviewer #5: Yes

4. Have the authors made all data underlying the findings in their manuscript fully available?

Reviewer #2: (No Response)

Reviewer #5: Yes

5. Is the manuscript presented in an intelligible fashion and written in standard English?

Reviewer #2: (No Response)

Reviewer #5: Yes

6. Review Comments to the Author

Reviewer #2: (No Response)

Reviewer #5: There are still some limitations in the study results and in the study design, which are also justified by the authors. The manuscript has gained significantly in quality through revision. The study is now described in a comprehensible way and the data and results support the authors' conclusion. However, the fact that the course of recovery may be more impaired after more severe pneumonia is not necessarily a surprise, but underscores the importance of medical follow-up.

7. PLOS authors have the option to publish the peer review history of their article (what does this mean?). If published, this will include your full peer review and any attached files.

Reviewer #2: No

Reviewer #5: No

---

## [Editor Report · Acceptance letter]

25 Aug 2021

PONE-D-21-11081R1 

Pulmonary function and six-minute-walk test in patients after recovery from COVID-19: A prospective cohort study 

Dear Dr. Wongsinin:

I'm pleased to inform you that your manuscript has been deemed suitable for publication in PLOS ONE. Congratulations! Your manuscript is now with our production department. 

Kind regards, 

on behalf of

Dr. Tai-Heng Chen 

Academic Editor

PLOS ONE